# Implication of Opioid Receptors in the Antihypertensive Effect of a Novel Chicken Foot-Derived Peptide

**DOI:** 10.3390/biom10070992

**Published:** 2020-07-02

**Authors:** Anna Mas-Capdevila, Lisard Iglesias-Carres, Anna Arola-Arnal, Gerard Aragonès, Begoña Muguerza, Francisca Isabel Bravo

**Affiliations:** 1Nutrigenomics Research Group, Department of Biochemistry and Biotechnology, Universitat Rovira i Virgili, 43007 Tarragona, Spain; anna.mas@urv.cat (A.M.-C.); lisard.iglesias@urv.cat (L.I.-C.); anna.arola@urv.cat (A.A.-A.); gerard.aragones@urv.cat (G.A.); franciscaisabel.bravo@urv.cat (F.I.B.); 2EURECAT-Technology Centre of Catalonia, Technological Unit of Nutrition and Health, 43204 Reus, Spain

**Keywords:** bioactive peptides, blood pressure, Caco-2 cells, naloxone, nitric oxide, UHPLC-MS/MS, UHPLC-HRMS

## Abstract

The peptide AVFQHNCQE demonstrated to produce nitric oxide-mediated antihypertensive effect. This study investigates the bioavailability and the opioid-like activity of this peptide after its oral administration. For this purpose, in silico and in vitro approaches were used to study the peptide susceptibility to GI digestion. In addition, AVFQHNCQE absorption was studied both in vitro by using Caco-2 cell monolayers and in vivo evaluating peptide presence in plasma from Wistar rats by ultra-high performance liquid chromatography-tandem mass spectrometry (UHPLC-MS/MS) and by ultra-high performance liquid chromatography-high resolution mass spectrometry (UHPLC-HRMS). Both in vivo and in vitro experiments demonstrated that peptide AVFQHNCQE was not absorbed. Thus, the potential involvement of opioid receptors in the BP-lowering effect of AVFQHNCQE was studied in the presence of opioid receptors-antagonist Naloxone. No changes in blood pressure were recorded in rats administered Naloxone, demonstrating that AVFQHNCQE antihypertensive effect is mediated through its interaction with opioid receptors. AVFQHNCQE opioid-like activity would clarify the antihypertensive properties of AVFQHNCQE despite its lack of absorption.

## 1. Introduction

Bioactive peptides are specific protein fragments released by limited proteolysis of their specific precursor proteins, which present additional biological activities over and above their expected nutritional [1,2]. Different health benefits have been reported for these peptides, including antioxidant, antithrombotic, antimicrobial, opioid, and anticancer activities [3]. Nevertheless, one of the most important biological properties attributed to bioactive peptides is their antihypertensive effects [4]. These peptides are usually obtained and selected by their capacity to inhibit angiotensin-converting enzyme (ACE), key enzyme in blood pressure (BP) regulation [5]. Hypertension (HTN) is a global public health problem [6] and although its medical treatment is well established, it can present side-effects in some patients [7]. Thus, the use of antihypertensive peptides as an alternative for HTN prevention and treatment is receiving interest even though these inhibitory peptides present lower ACE inhibitory (ACEI) activity than the antihypertensive drugs [4].

Other mechanisms than the inhibition of ACE could be involved in the bioactive peptide BP-lowering effect, including an improvement in endothelial function by the activation of nitric oxide (NO) pathway [8]. In this sense, NO has also been reported to mediate the endothelium-dependent vasorelaxation effects of different antihypertensive peptides [8]. The antihypertensive peptides that mediate its effect through NO could enhance NO production by the activation of endothelin NO synthase by phosphorylation at Ser-1177 residue consistent [9]. Additionally, some bioactive peptides also present antioxidant effects, which in turn would contribute to enhance NO bioavailability and to improve endothelium functionality by reducing radical oxygen species, that are known to be NO scavengers [10].

The peptide in vivo physiological effects depend on their ability to reach the target sites in an active form [11]. In this context, gastrointestinal (GI) digestion plays a key role in the formation and degradation of bioactive peptides [12]. Once ingested, peptides might be subjected to hydrolysis by different GI enzymes such as pepsin, trypsin, chymotrypsin, and peptidases from the surface of epithelial cells, which could result in the release of different amino acid sequences [13]. In addition to digestion, absorption may also condition peptide bioactivity. In this sense, peptides are absorbed depending on their size and structure. Di- and tri-peptides are frequently transported through PepT1 peptide transporter [14], while larger peptides can be absorbed by passive diffusion [15]. Moreover, water-soluble large peptides can cross via the tight junctions between cells and highly lipid-soluble peptides appear to be able to diffuse via the transcellular route [14].

However, some bioactive peptides might mediate their physiological effects through its binding to receptors present in the intestinal wall, implying that their absorption is not required [13]. It has been demonstrated that some opioid peptides are not absorbed and can produce their biological effects by the interaction with opioid receptors (μ, δ, and κ), which have been described in several tissues including in the GI tract [15,16]. These opioid-like peptides are atypical opioid ligands that are released from the hydrolysis of food and present N-terminal sequences that are different from those of the typical opioid peptides such as endorphins. Some common structural features of opioid peptides are the presence of a tyrosine residue at the amino-terminal end, and an aromatic residue, such as phenylalanine or tyrosine, in the third or fourth position. These structural motives seem to be necessary for binding to the opioid receptors, however, there is still no consensus about critical opioid-like structures [17]. In this regard, some opioid peptides have demonstrated to regulate BP and produce its BP-lowering effect by binding to opioid receptors present in the GI tract [18]. Nurminen et al. [19] evidenced the involvement of opioid receptors in the antihypertensive effect of the peptide α-lactorphin (YGLF) as its BP-lowering effect was abolished by the administration of the non-selective opioid receptor antagonist, Naloxone [20]. Furthermore, Sipola et al. [21] demonstrated α-lactorphin produced its opioid-mediated antihypertensive effect through NO release.

AVFQHNCQE is a nonapeptide initially identify in the chicken foot hydrolysate Hpp11, which exhibited antihypertensive properties after its oral administration to spontaneously hypertensive rats (SHR) [22]. In particular, AVFQHNCQE produced a NO-mediated antihypertensive effect and it was demonstrated that enhanced NO bioavailability through its antioxidant effect and by improving the endothelium functionality when administered to the hypertensive animals [23]. However, AVFQHNCQE vulnerability to GI enzymes and posterior absorption has not been studied yet. Thus, this study aimed to investigate the bioavailability of the antihypertensive peptide AVFQHNCQE after its oral administration. For this purpose, in silico and in vitro approaches were used to study the peptide susceptibility to GI digestion. In addition, AVFQHNCQE absorption was studied both in vitro by using Caco-2 cell monolayers and in vivo evaluating peptide presence in plasma from Wistar rats [14]. Finally, the potential involvement of opioid receptors in the BP-lowering effect of AVFQHNCQE was studied in the presence of opioid receptors-antagonist Naloxone.

## 2. Materials and Methods

### 2.1. Materials

The peptide AVFQHNCQE was synthesized by CASLO Aps. (Kongens Lyngby, Denmark) and its purity was 98.94%. Captopril was provided by Santa Cruz Biotechnology (Dallas, TX, United States). Pepsin, bile salts mixture, trypsin, α-chymotrypsin, porcine pancreatic lipase, colipase, trifluoroacetic acid (TFA), formic acid (FA) LC-MS grade, and bovine serum albumin (BSA) were obtained from Sigma-Aldrich (Madrid, Spain). Acetonitrile and methanol LC-MS grade were purchased from Merck (Darmstadt, Germany). The synthetic isotopically labeled peptide IFV*TGQDYNDK (V* = Val-13C5,15N) was obtained from Biomatik (Ontario, Canada) and used as internal standard (IS) for MS experiments. Chicken foot hydrolysate Hpp11 was obtained by our group following the procedure described by Bravo et al. [22,24]. All other chemical solvents used were of analytical grade.

### 2.2. In Silico and In Vitro Simulated Digestion

In silico simulated peptide digestion was carried out using the programs ExPASy PeptideCutter, available at http://web.expasy.org/peptide_cutter. This approach can predict the hydrolysis of a protein sequence obtained from a protein database using the known enzymatic cleavage sites. The digestive enzymes pepsin, chymotrypsin, trypsin, lipase, and colipase were used to predict the protein fragments generated in GI digestion of AVFQHNCQE.

For the in vitro simulated peptide GI digestion, a two-stage hydrolysis process was carried out according to Martos et al. [25]. AVFQHNCQE was dissolved in simulated gastric fluid (35 mM NaCl at pH 2 for 15 min) and posteriorly subjected to digestion by porcine pepsin (E.C. 3.4.23.1, 3440 units/mg) at an enzyme/substrate ratio of 1: 20 *w*/*w* at 37 °C for 60 min. Gastric digestion with pepsin was stopped by adding 1 M NaHCO3 and pH was adjusted to 7.0 with NaOH 1 M (final protein concentration of 5 mg/mL). Aliquots of this gastric digest were collected (G60) and stored at −20 °C until analysis. Duodenal digestions were performed by using, as the starting material, the obtained gastric digests, with the addition of 1 M CaCl2, 0.25 M bis-Tris pH 6.5, 0.25 M bis-Tris pH 6.5, and a 0.125 M bile salts mixture containing equimolar quantities of sodium taurocholate and glycodeoxycholic acid. Posteriorly, the duodenal enzymes trypsin (EC 232-650-8, 10100 BAEE units/mg protein), α-chymotrypsin (EC 232-671-2; 55 units/mg protein), porcine pancreatic lipase (EC 232-619-9) and colipase (EC 259-490-1), prepared in 35 mM NaCl adjusted to pH 7, were added to the solution. Duodenal digestion was carried out at pH 7, 37 °C during 60 (D60). Then, enzymes were inactivated by heating at 95 °C for 10 min in a water bath, followed by cooling to room temperature. Triplicates of aliquots at each time point were collected and stored at −20 °C until analysis.

### 2.3. Peptide Bioavailability Studies

#### 2.3.1. In Vivo Experiment

Male Wistar rats (17–20 weeks-old, weighing between 230 and 250 g) were obtained from Charles River Laboratories (Barcelona, Spain). Animals were maintained in pairs, at 22 °C with light/dark cycles of 12 h and were fed standard chow diet (AO4, Panlab, Barcelona, Spain) ad libitum during all the experiment. Animals were randomly divided and administered tap water (control group, *n* = 6) or 10 mg/kg body weight (bw) peptide (AVFQHNCQE group, *n* = 6) by gastric intubation after starvation of 12 h. The total volume of water or AVFQHNCQE orally administered to the rats was 1 mL. Blood samples were collected via saphenous vein extraction by use of heparin vials (Starsted, Barcelona, Spain) before the water or peptide administration and 30 min and 60 min post-administration. Plasma samples were obtained by blood centrifugation (2000× *g*, 15 min, 4 °C) and were pooled (*n* = 6 per treatment group) to perform ultra-high performance liquid chromatography-tandem mass spectrometry (UHPLC-MS/MS) analysis. Plasma was stored at −80 °C until the analysis.

#### 2.3.2. In Vitro Experiment

Caco-2 cells were obtained from Sigma-Aldrich and were grown Dulbecco’s modified minimum essential medium (DMEM), supplemented with 20% fetal bovine serum (FBS), 2 mmol/mL L-glutamine, 100 U/mL penicillin and 100 μg/mL streptomycin at 37 °C in and 5% CO_2_. Cells were seeded onto Transwell inserts permeable membrane support (0.4 µm pore, 24 mm diameter, 4.7 cm^2^ grown surface from Merck Co., Bedford, MA, USA) placed in six-well plates. The seeding density was 12,000 cell/cm^2^. The medium was replaced every 2–3 days and cells were growing for 21 days. On the 21st day of the procedure, the integrity of the Caco-2 monolayer was assessed by the measurement of transepithelial electrical resistance (TEER) using a Millicell ERS-2 voltammeter (EMD Millipore, Darmstadt, Germany) selecting the ones with TEER values higher than 200 Ω/cm^2^.

For the transport study, Caco-2 cells, maintained in DMEM, were gently rinsed with Hank’s balanced salt solution (HBSS) and equilibrated for 20 min at 37 °C prior to the transport study. Then, it was evaluated the transepithelial transport of the peptide AVFQHNCQE (1 mM) dissolved in HBSS and the digests obtained in the in vitro simulated digestion (G60 and D60). AVFQHNCQE or the peptide digests G60 and D60 were added to the apical (AP) chamber and were incubated for 1 h at 37 °C. After this time, triplicates of the AP chamber content and the basolateral (BL) chamber content were taken and stored at −80 °C until ultra-high performance liquid chromatography-high resolution mass spectrometry (UHPLC-HRMS) analyses.

### 2.4. Peptide Analysis by UHPLC-MS/MS and UHPLC-HRMS

#### 2.4.1. Optimization of Peptide Extraction from Plasma

To efficiently extract the peptide or peptide fragments from plasma after AVFQHNCQE administration in Wistar rats, three different extraction methods were assayed.

**Solid-phase extraction (SPE)**: Plasma samples (150 µL) were mixed with 25 µL of IFV*TGQDYNDK (IS) (10 ppm) and 800 µL of H_2_O (1% TFA). Then, these solutions were heated at 95 °C for 2 min to denature plasma proteins. After cooling down, samples were centrifuged (15,000 rpm, 15 min, 4 °C) and loaded to Oasis HLB (30 mg, 1 mL) cartridges (Waters, Barcelona, Spain), which were sequentially pre-conditioned with 1 mL acetonitrile: water 0.1% FA (75:25, *v*/*v*) and 1 mL water 0.1% FA. Loaded cartridges were washed with 1 mL water 0.1% FA and dried. Retained peptides were eluted with 2 sequential additions of 250 µL acetonitrile: water 0.1% FA (75:25, *v*/*v*). Eluted samples were dried in a speed-vac concentrator (Thermo Fisher, Waltham, Massachusetts), USA), reconstituted in 100 µL of water (0.1 % FA and 0.1 % BSA), and analyzed by UHPLC-MS/MS. The extraction was performed in triplicate.

**Protein precipitation by TFA:** 150 µL of plasma were mixed with 25 µL of IS (10 ppm) and 40 µL of water (10% TFA) and heated for 2 min at 95 °C to precipitate plasma proteins. After cooling down, samples were centrifuged (15,000 rpm, 15 min, 4 °C) and supernatants were analyzed by UHPLC-MS/MS. The extraction was performed in triplicate.

**Protein precipitation by TFA and solid-phase extraction (SPE):** Plasma samples (150 µL) were mixed with 25 µL of IS (10 ppm) and 40 µL of water (10% TFA) and heated at 95 °C for 2 min. After cooling down, samples were centrifuged (15,000 rpm, 15 min, 4 °C) and supernatants were purified using Oasis HLB (30 mg, 1 mL) cartridges (Waters) with the same procedure described before and analyzed by UHPLC-MS/MS. The extraction was performed in triplicate.

#### 2.4.2. Peptide Extractions from In Vitro Digestions and Caco-2 Monolayers

Samples obtained from in vitro peptide simulated digestion and Caco-2 monolayers were diluted 10 times with water (0.1% FA) and centrifuged (15,000 rpm, 15 min, 4 °C). Supernatants were directly analyzed by UHPLC-HRMS to avoid the loss of polar peptide fragments. Extensive clean-up or pre-concentration procedures were not necessary due to the high concentration of peptide and low matrix complexity than plasma samples.

#### 2.4.3. Analysis by UHPLC-MS/MS and UHPLC-HRMS

The obtained purified solutions from plasma and peptide digests G60 and D60 and from AP and BL chambers in Caco-2 experiment were analyzed using a UHPLC 1290 Infinity II Series coupled to a QqQ 6490 Series (Agilent Technologies, Santa Clara, CA, USA) for tandem mass experiments or a qTOF 6550 Series (Agilent Technologies) for high-resolution mass spectrometry experiments. Chromatographic separation of peptide or their fragments were performed using a Kinetex 2.6 µm EVO C18 column (150 × 2.1 mm, 2.7 µm) from (Phenomenex, Torrance, USA) as stationary phase and water containing 0.1% formic acid (solvent A) and acetonitrile (solvent B) as mobile phase. The separation was performed in gradient elution as follows: isocratic; 0–0.5 min, 2 % B; 0.5–1 min, 2–6% B; 1–5 min, 6–15% B; 5–6 min, 15–35% B; 6–10 min, 35–98% B; 10–12 min, 98% B isocratic; 12–12.10, 98–2% B. For all runs, 10 μL of the sample (4 °C) was injected, and the flow rate was 0.350 mL/min and the column temperature was 25 °C.

Peptide ionization was performed in positive electrospray ionization (ESI) and source parameters were as follows and the same for both mass spectrometer instruments: gas temperature 150 °C; gas flow 13 L/min, sheath gas temperature 250 °C; sheath gas flow 11 L/min; nebulizer pressure 40 psi capillary voltage 3000 V, and nozzle voltage 500 V. For plasma samples, QqQ instrument was used to quantify the AVFQHNCQE peptide because a low concentration was expected. The mass spectrometer was operated in multiple reaction monitoring (MRM) mode and optimized transitions were 538.4 > 928.9 (10 eV) for quantification and 538.4 > 758.5 (10 eV) and 538.4 > 453.4 (10 eV) for confirmation (RT = 4.25 min) of AVFQHNCQE and 653.4 > 940.8 (20 eV) for quantification and 653.4 > 1046.0 (20 ev) and 653.4 > 839.7 (25 eV) for confirmation (CE = 20 eV, RT = 6.15 min) of IFV*TGQDYNDK (IS). Peptide quantification was performed by an internal standard calibration curve using the pure standard of AVFQHNCQE peptide with a limit of detection of 0.05 ppb and limit of quantification of 0.130 ppb in plasma and a linear range up to 166 ppb (r2 > 0.9999).

For in vitro simulated digestion experiments and Caco-2 monolayer samples, qTOF instrument was used to investigate the presence of unknown peptide fragments and previously in silico predicted peptide fragments by means of its accurate mass. Peptide and its fragments were semi-quantified using the chromatographic peak area observed in the extracted ion chromatograms using a 20 ppm mass window.

### 2.5. Evaluation of the Opioid-Like Activity of AVFQHNCQE in SHR

Male, 17–20-week-old, SHR, weighing 286 ± 4 g were used in this study. All these animals were obtained from Charles River Laboratories Spain. The SHR were housed at a temperature of 23 °C with 12 h light/dark cycles and had free access to tap water and a standard diet (A04 Panlab). Before the experiments, rats were submitted to a training period of 10 days to guarantee the reliability of the measurements. To evaluate if AVFQHNCQE presented an opioid-mediated effect, SHR received via subcutaneous 100 µL of Naloxone (3 mg/kg bw), and afterward a single oral dose (1 mL) of AVFQHNCQE (10 mg/kg bw). Additionally, water, Captopril (50 mg/kg bw), known antihypertensive drug, and chicken foot hydrolysate Hpp11 (55 mg/kg bw), were administered as controls. Captopril and Hpp11 can inhibit ACE in vivo. All treatments were administered by oral gavage between 8:00 and 9:00 h a.m. The total volume orally administered to the rats for all treatments was 1 mL. Systolic blood pressure (SBP) was recorded in the rats by the tail-cuff method [26] before and 2, 4, and 6 h post-administration. All measurements were taken by the same person in the same peaceful environment.

The Animal Ethics Committee of University Rovira i Virgili approved all procedures (reference number 8868 by Generalitat de Catalunya). All the above-mentioned experiments were performed as authorized (European Directive 86/609/CEE and Royal Decree 223/1988 of the Spanish Ministry of Agriculture, Fisheries and Food, Madrid, Spain).

### 2.6. Statistical Analysis

Results from transepithelial transport studies are presented as means ± standard error means (SEM). Means were compared using Student’s t-test. The results for the experiments investigating the participation of opioid receptors in the AVFQHNCQE, Captopril and Hpp11 antihypertensive effect were expressed as the mean ± SEM of six animals per group (heterogeneity was tested by Levene’s test) and were analyzed by one-way ANOVA (Tukey’s test) using IBM SPSS Statistics 20.0 software. Outliers were identified and eliminated by using Grubbs’ test. Differences between groups were considered significant when *p* < 0.05.

## 3. Results and Discussion

Food proteins contain specific peptide sequences that are inactive as long as they remain bonded to other amino acids within the primary protein structure, which might be released by protein hydrolysis [27]. These peptides can present a wide range of bioactivities including ACEI and antihypertensive activity [27]. In this sense, it has been demonstrated that the hydrolysis of chicken foot proteins is a good strategy for the obtaining of bioactive peptides able to reduce BP [22,24]. The NO-mediated antihypertensive effect of the peptide AVFQHNCQE (10 mg/kg bw), identified initially in the chicken foot hydrolysate Hpp11, has been previously demonstrated in SHR [22,23]. Nevertheless, it is unknown if this peptide sequence is susceptible to GI digestion, resulting in new peptide fragments which might be the responsibility of AVFQHNCQE bioactivity. Therefore, the peptide fragmentation under GI digestion of the antihypertensive peptide was investigated using in silico and in vitro approaches. Simulated in silico digestion considers the primary structure of peptides and the known cleavage specificity of the GI tract enzymes, predicting the peptide potential to release other shorter peptides sequences during digestion. Table 1 shows the amino acid sequences of peptides predicted to be released after AVFQHNCQE gastric and duodenal digestion based on the in silico digestion approach. This model predicted a peptide hydrolyzed only by pepsin and chymotrypsin. Trypsin was also evaluated showing no effects on peptide hydrolysis. Although the in silico approach is important for the identification of the potential bioactive amino acid sequences, the results of these studies have to be interpreted with caution, as this approach does not consider the tertiary structure of the proteins. However, based on the in silico prediction, it would appear that AVFQHNCQE could release different amino acid sequences, which could be involved in its antihypertensive effect.

Simulated in vitro gastrointestinal digestion model was used to investigate the releasing of the amino acid sequences previously predicted in the in silico approach. The peptide was sequentially digested under physiological conditions and the identification of amino acid sequences in samples obtained from both gastric and intestinal digestion were performed by UHPLC-HRMS (Table 2). When the results obtained by the in silico approach and in vitro approach were compared, it was observed that only two of the fourteen predicted peptide fragments from the in silico approach were identified in the in vitro digested peptide samples. These findings could be explained by the limitations presented by the in silico approach that does not take into consideration the tertiary structure of proteins and the physiological conditions during the gastrointestinal (temperature, pH). From the in vitro approach, it was observed in sample G60 the presence of the native peptide and also the presence of different peptide fragments. However, the most intense signal corresponded to the native peptide, AVFQHNCQE, indicating that this peptide could be resistant to gastric digestion. In this sense, Ruiz et al. reported for the peptides VRYL and KKYNVPQL, derived from Manchego cheese, their resistance to GI digestion, and their ability to reach their target organs as an intact form. Interestingly, KKYNVPQL, such as AVFQHNCQE, contains glutamine residue (Q) in the C-terminal, which is commonly presented in peptides resisting GI digestion hydrolysis [28]. Regarding the duodenal digests, the most intense signal after 60 min under simulated intestinal digestion (D60) corresponded to free amino acids. Taking into account that most antihypertensive peptides require to be absorbed in the intestine to produce their BP-lowering effect [11], absorption of the AVFQHNCQE or their potential derived fragments should produce before 60 min of intestinal digestion.

Once peptide fragmentation pattern during gastrointestinal digestion was predicted by using in silico and in vitro approaches it was aimed to explore the bioavailability of the entire peptide sequence or predicted peptide fragments in the circulation blood after its oral administration. Thus, an additional in vivo study was carried out in which the presence of the AVFQHNCQE or their fragments were evaluated by chromatographic analysis in Wistar rats plasma collected before and 30 and 60 min post-administration of AVFQHNCQE (10 mg/kg bw) or water. However, as no specific and optimized peptide extraction methodology exist for AVFQHNCQE using plasma samples, this procedure was optimized and validated to obtain the maximum recovery and reproducibility. Three different extraction methodologies (SPE, TFA, and SPE + TFA) were evaluated using samples spiked with 1 ppb of AVFQHNCQE peptide to determine recovery (%), repeatability (% RSD) and matrix effect (%). SPE method was demonstrated to be the most efficient method for the peptide extraction in plasma obtaining a recovery of 92.4% with good reproducibility (Appendix A) and a negligent matrix effect (less than 20%). Once the peptide extraction method was optimized, rat plasma at 0, 30, and 60 min after peptide (10 mg/kg bw) or water oral administration were subjected to SPE extraction and analyzed by UHPLC-MS/MS for AVFQHNCQE peptide quantification or, by UHPLC-HRMS for peptide fragment analysis. Interestingly, neither the peptide nor their potential fragments were identified in the plasma samples studied, indicating that none of them is absorbed.

To verify the lack of absorption of the peptide or their resultant fragments in the GI tract an additional study using Caco-2 cells was carried out. The native peptide AVFQHNCQE and the samples collected after the simulated gastric and duodenal digestions (G60 and D60, respectively) were incubated in Caco-2 monolayer for 1 h. The peptide and its resultant fragments were identified in the AP chamber from G60 samples, however, only 0.10–0.25% were transported to the BL chamber (Table 3). Therefore, the presence of native AVFQHNCQE in the AP chamber after 1 h of incubation and the absence of the peptide or their fragments in the BL chamber demonstrated that it was not absorbed through the intestinal monolayer. Similarly, Miguel et al. reported for the antihypertensive peptide RADHP the lack of transported through Caco-2 monolayer [12].

The absence of the peptide AVFQHNCQE or their potential derived fragments in the mass spectrometry data from the plasma or BL chamber samples suggest that AVFQHNCQE can exert its antihypertensive effect without requiring to be absorbed, by interacting with receptors present in the intestinal tract, as has been previously reported for other large peptides [15]. In this sense, it has been demonstrated that some antihypertensive peptides that also present opioid activity do not require to be absorbed to induce BP reduction. These opioid peptides induce antihypertensive effect by their interaction with the opioid receptors present in the GI tract [29]. The mechanism of action proposed driven by the stimulation of these opioid receptors is the subsequent NO release causing vasodilation [30,31,32,33]. Therefore, considering that AVFQHNCQE was not absorbed and its antihypertensive effect was mediated by NO [23], it was investigated the participation of opioid receptors in its effect on blood pressure. Thus, the antihypertensive properties of AVFQHNCQE were studied in presence of Naloxone, which is an opioid receptor antagonist [34]. Figure 1, Figure 2 and Figure 3 show the changes in SBP 6 h after oral administration of the peptide, the chicken foot hydrolysate Hpp11 and Captopril in rats previously administered via subcutaneous Naloxone or saline. The results of this study showed that the antihypertensive effect of AVFQHNCQE was completely abolished in naloxone-treated rats while in the control rats, administered saline, it was observed −31 ± 2 mmHg of SBP decrease when compared to water group (Figure 1). Similarly to our results, Nurminen et al. reported for the milk-derived peptide α-lactorphin opioid-mediated antihypertensive effect [19]. It was also demonstrated that the interaction of the milk-derived peptide α-lactorphin with opioid receptors resulted in a NO-mediated antihypertensive effect [20]. Therefore, considering the present findings, AVFQHNCQE produced its NO-mediated antihypertensive effect through the interaction with opioid receptors, clarifying the reason why this peptide does not require to be absorbed to produce its physiological effect.

Interestingly, Hpp11 chicken foot hydrolysate, in which initially identified the peptide AVFQHNCQE, was able to reduce BP in both groups, in the naloxone-treated group and saline-treated group indicating that its BP-lowering effect was not mediated through the interaction with opioid receptors (Figure 2). According to this, Hpp11 was demonstrated to produce its antihypertensive effect by reducing ACE activity [24]. According to this, Captopril, an ACEI antihypertensive drug [35], was also able to reduce BP in rats treated with Naloxone and in rats treated with saline, confirming that as an ACE inhibitor, its effect is not mediated by opioid receptor and it is absorbed and transported to reach the cardiovascular system and to inhibit ACE (Figure 3) [36].

## 4. Conclusions

In the present study, it was demonstrated that AVFQHNCQE showed high resistance to gastric digestion, but it was mainly hydrolyzed into free amino acid residues after 60 min in the duodenum. In addition, this peptide was not absorbed through intestinal epithelium, indicating that AVFQHNCQE does not require its absorption to produce its BP-lowering effect. Indeed, it was demonstrated that the peptide produces its NO-mediated antihypertensive effect through their interaction with opioid receptors. Future studies are required to demonstrate their antihypertensive effect in humans. Nevertheless, the results presented in this work that includes the mechanistic study about the way of action of AVFQHNCQE are relevant because they open doors to the use of this chicken-foot derived peptide to formulate antihypertensive functional foods or nutraceuticals that could allow to treat prehypertension or to prevent the development of hypertension.

## Figures and Tables

**Figure 1 biomolecules-10-00992-f001:**
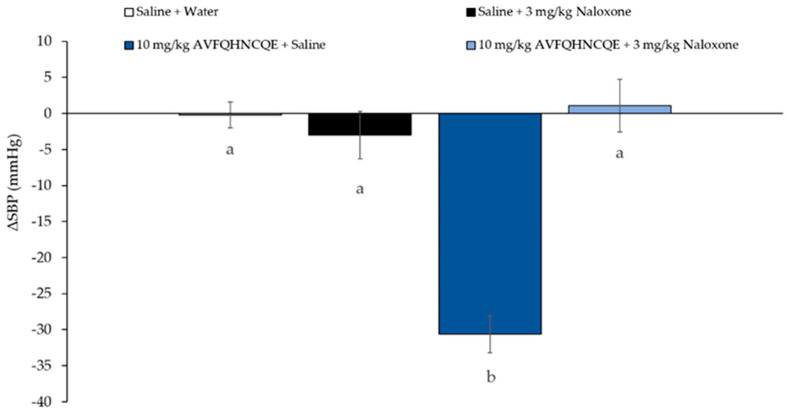
Changes in systolic blood pressure (SBP) caused in spontaneously hypertensive rats after 6 h post-administration of the different treatments: oral administered water + intraperitoneal injected saline, oral administered water saline + intraperitoneal injected 3 mg/kg bw Naloxone, oral administered water 10 mg/kg bw AVFQHNCQE + intraperitoneal injected saline or oral administered water 10 mg/kg bw AVFQHNCQE + intraperitoneal injected 3 mg/kg bw Naloxone. Data are expressed as mean ± standard errors. The experimental groups always had six animals each. Different letters represent statistically significant differences (*p* < 0.05). P was estimated by one-way ANOVA.

**Figure 2 biomolecules-10-00992-f002:**
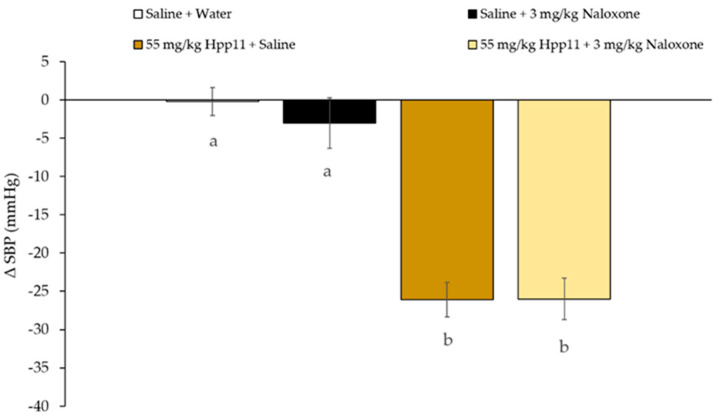
Changes in systolic blood pressure (SBP) caused in spontaneously hypertensive rats after 6 h post-administration of the different treatments: oral administered water + intraperitoneal injected saline, oral administered saline + intraperitoneal injected 3 mg/kg bw Naloxone, oral administered 55 mg/kg Hpp11 + intraperitoneal injected saline or oral administered 55 mg/kg bw Hpp11 + intraperitoneal injected 3 mg/kg bw Naloxone. Data are expressed as mean ± standard errors. The experimental groups always had six animals each. Different letters represent statistically significant differences (*p* < 0.05). P was estimated by one-way ANOVA.

**Figure 3 biomolecules-10-00992-f003:**
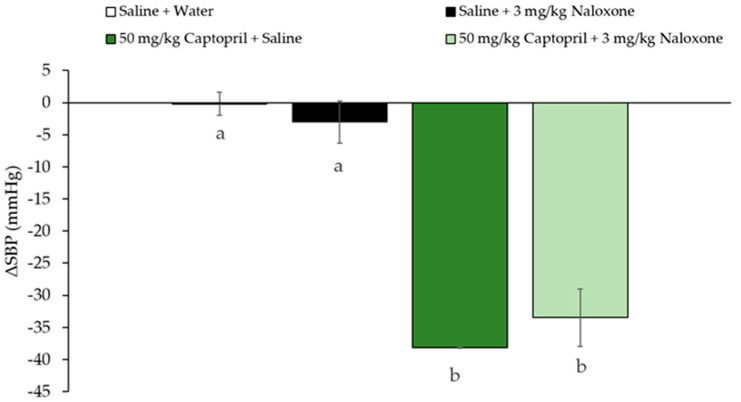
Changes in systolic blood pressure (SBP) caused in spontaneously hypertensive rats after 6 h post-administration of the different treatments: oral administered water + intraperitoneal injected saline, orally administered saline + intraperitoneal injected 3 mg/kg bw Naloxone, oral administered 50 mg/kg bw Captopril + intraperitoneal injected saline or orally administered 50 mg/kg Captopril + intraperitoneal injected 3 mg/kg bw Naloxone. Data are expressed as mean ± standard errors. The experimental groups always had six animals each. Different letters represent statistically significant differences (*p* < 0.05). P was estimated by one-way ANOVA.

**Table 1 biomolecules-10-00992-t001:** In silico simulated digestion of the peptide AVFQHNCQE.

Original Sequence	Enzyme	Digestion Stage	Final Sequence
AVFQHNCQE	Pepsin	Gastric	VFQHNCQE
AVFQHNCQE	Pepsin	Gastric	AV
AVFQHNCQE	Pepsin	Gastric	FQHNCQE
VFQHNCQE	Chymotrypsin	Duodenal	VF
VFQHNCQE	Chymotrypsin	Duodenal	QHNCQE
QHNCQE	Chymotrypsin	Duodenal	VFQH
QHNCQE	Chymotrypsin	Duodenal	NCQE
VFQHNCQE	Chymotrypsin	Duodenal	QHNCQE
VFQHNCQE	Chymotrypsin	Duodenal	FQHNCQE
FQHNCQE	Chymotrypsin	Duodenal	QHNCQE
FQHNCQE	Chymotrypsin	Duodenal	FQH
FQHNCQE	Chymotrypsin	Duodenal	NCQE
FQHNCQE	Chymotrypsin	Duodenal	QH
FQHNCQE	Chymotrypsin	Duodenal	NCQE

**Table 2 biomolecules-10-00992-t002:** Identified sequences from the in vitro digestion of the peptide AVFQHNCQE.

Peptide Sequence ^a^	*m*/*z*^b^	Mass Monoisotopic (Da)	RT ^c^	Charge	Area G60 ^d^	Area D60 ^e^
AVFQHNCQE	538.233	1074.451	4.86	2	5,236,186	13,652
AVFQHNCQE(S-S)AVFQHNCQE	537.729	2146.887	5.24	4	605,812	-
AVF	336.191	335.183	5.83	1	200,712	-
AVFQHNC	409.682	817.350	4.72	2	220,051	-
AVFQHN	358.179	714.343	4.24	2	685,482	-
AVFQHNCQE(S-S)QHNCQE	458.436	1829.715	4.11	4	136,056	-
AVFQH	301.157	600.300	4.34	2	155,533	-
AVFQHNCQE(S-S)CQE	484.526	1450.554	4.25	3	93,694	-
QHNCQE	379.647	757.278	1.23	2	288,536	335,871
NCQE	493.169	492.161	1.27	1	-	57,009

^a^ Amino acid residues are designated using their one letter codes; ^b^
*m*/*z* = mass-to-charge; ^c^ RT = retention time; ^d^ G60 = samples from 60 min of gastric digestion; ^e^ D60 = samples from 60 min of duodenal digestion.

**Table 3 biomolecules-10-00992-t003:** Transport of peptide and fragments from G60 through Caco-2 monolayer.

Peptide Sequences ^a^	*m/z* ^b^	Mass Monoisotopic (Da)	RT ^c^	Charge	G60 % Transport ^d^
AVFQHNCQE	538.233	1074.451	4.86	2	0.25
AVFQHNCQE(S-S)AVFQHNCQE	537.729	2146.887	5.24	4	0.11
AVF	336.191	335.183	5.83	1	0.15
AVFQHNC	409.682	817.350	4.72	2	0.60
AVFQHN	358.179	714.343	4.24	2	0.10
AVFQHNCQE(S-S)QHNCQE	458.436	1829.715	4.11	4	0.15
AVFQH	301.157	600.300	4.34	2	0.17
AVFQHNCQE(S-S)CQE	484.526	1450.554	4.25	3	0.15
QHNCQE	379.647	757.278	1.23	2	N.D
NCQE	493.169	492.161	1.27	1	N.D

^a^ Amino acid residues are designated using their one-letter codes; ^b^
*m/z* = mass-to-charge; ^c^ RT = retention time; ^d^ % transport: percentage of the peptide or peptide fragments content found in the basolateral chamber in comparison to the apical chamber content. D60 = samples from 60 min of duodenal digestion.

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
