# Peer review of "Implication of Opioid Receptors in the Antihypertensive Effect of a Novel Chicken Foot-Derived Peptide"

_biomolecules, 2020, doi:10.3390/biom10070992_

Round 1

Reviewer 1 Report

This manuscript clearly demonstrated that the hypertensive effect of oral nonapeptide AVFQHNCQE in an opioid receptor antagonist-reversible manner in rat. Furthermore, authors showed that 14 possible digested shorter peptides to be released after AVFQHNCQE gastric and duodenal digestion based on the in silico digestion approach. In addition, authors demonstrated that none of the 10 detected digested peptides or the original peptide AVFQHNCQE were absorbed by using intestinal Caco-2 monolayer in vitro model. Finally, they demonstrated that the hypertensive effect of oral AVFQHNCQE was significantly and completely reversed by systemic Naloxone treatment.

            The results of this study may include beneficial information to understand the mechanism of the anti-hypertensive effect of AVFQHNCQE, but I found several problems as following.

Specific comments

  1. Among the 14 digested peptides found by using in silico simulated digestion model (Table1), only 2 peptides are experimentally found (QHNCQE and NCQE) by using in vitro digestion (Table2). Authors should discuss on such differences of the results between in silico and in vitro.
  2. Authors describes the absorption (or bioavailability) of the digested peptides using in vitro model (Caco-2 monolayer cells). It would be more supportive if the authors determine/show the blood concentration of the peptides after oral administration (in vivo).
  3. In the combination in vivo experiment (Figures 1, 2 and 3), authors injected naloxone systemically (i.p.) before the oral administration of the reagents (AVFQHNCQE, Hpp11, or Captopril). Since naloxone will reach  everywhere in the body after systemic injection, it would be better to inject naloxone orally to show and expect the direct interaction between the peptides and the opioid receptors at the intestine before absorption.
  4. The mechanism of the intestinal opioid-receptor mediated hypertension is not clear in this manuscript. Authors should add the description in the discussion. Will the peptides can bind to the opioid receptors?
  5. If possible, authors should demonstrate the release of nitric oxide by the oral nonapeptide and the involvement of opioid receptors in the mechanism.

Author Response

Manuscript ID: biomolecules-839728

Dear reviewer, Please find the point-by-point response to each of your comments with the description of the changes made in the manuscript. Thank you very much for your attention reconsidering our manuscript for its publication in Biomolecules.Yours sincerely,Begoña Muguerza, corresponding author.

REVIEWER 1

This manuscript clearly demonstrated that the hypertensive effect of oral nonapeptide AVFQHNCQE in an opioid receptor antagonist-reversible manner in rat. Furthermore, authors showed that 14 possible digested shorter peptides to be released after AVFQHNCQE gastric and duodenal digestion based on the in silico digestion approach. In addition, authors demonstrated that none of the 10 detected digested peptides or the original peptide AVFQHNCQE were absorbed by using intestinal Caco-2 monolayer in vitro model. Finally, they demonstrated that the hypertensive effect of oral AVFQHNCQE was significantly and completely reversed by systemic Naloxone treatment.

 The results of this study may include beneficial information to understand the mechanism of the anti-hypertensive effect of AVFQHNCQE, but I found several problems as following.

Specific comments

Comment 1: Among the 14 digested peptides found by using in silico simulated digestion model (Table1), only 2 peptides are experimentally found (QHNCQE and NCQE) by using in vitro digestion (Table2). Authors should discuss on such differences of the results between in silico and in vitro.

Response:

As well pointed by the reviewer, our findings demonstrate that results of in silico and in vitro approaches in terms of simulated gastrointestinal digestion could differ between one another. Similar results were also observed by other authors.

Studying digestive processes is challenging due to its complexity of the digestive process, the complexity nature of food matrixes and the limitations of currently available techniques. In silico models simulate digestive processes based on the amino acid sequence of the intact protein and knowledge about the specificity of proteases in the gastrointestinal tract. Being an in silico model, it cannot be concluded with certainty that the purported bioactive peptides will be generated after the in vitro and the actual and in vivo gastrointestinal digestion. In silico model predicts the possible protein fragmentation pattern or amino acid pattern release considering the cleavage sites of different gastrointestinal enzymes. In the present study, in silico model predict that only pepsin and chymotrypsin were able to cleavage the studied peptide sequence. Although in silico approach is important for the identification of the amino acid sequences that could result after the digestion, the results of these studies have to be interpreted with caution, since this approach does not consider the tertiary structure of the proteins. In addition, in vitro simulated digestion followed a more complex procedure which included pepsin, trypsin, chymotrypsin, lipase and colipase as gastric and duodenal enzyme. Also, this procedure took into consideration environmental conditions during the gastrointestinal digestion including gastric and duodenal fluids, temperature and pH. All these parameters that are not considered in the in silico approach could explain the differences observed between the results obtained by in silico and in vitro approach.

As suggested by the reviewer we discuss further the differences observed between in silico and in vitro approach by including the following information (lines 260-264).

“When compared the results obtained by in silico approach and in vitro approach it was observed that only two of the fourteen predicted peptide fragments from the in silico approach were identified in the in vitro digested peptide samples. These findings could be explained by the limitations presented by the in silico approach that does not take into consideration the tertiary structure of proteins and the physiological conditions during the gastrointestinal (temperature, pH). From the in vitro approach it was observed in sample G60 the presence of the native peptide and also the presence of different peptide fragments.”

Comment 2: Authors describes the absorption (or bioavailability) of the digested peptides using in vitro model (Caco-2 monolayer cells). It would be more supportive if the authors determine/show the blood concentration of the peptides after oral administration (in vivo).

Response:

We agree with the referee about the importance to also investigate the presence of peptide and peptide fragments in vivo and that is why we also included in the manuscript the results of peptide or peptide fragments bioavailability in plasma after 30 and 60 minutes after its oral administration. However, we included the following sentence to clarify the order of the experiments followed in the present manuscript (lines 280-284).

“Once peptide fragmentation pattern during gastrointestinal digestion was predicted by using in silico and in vitro approaches it was aimed to explore the bioavailability of the entire peptide sequence or predicted peptide fragments in the circulation blood after its oral administration”.

Comment 3: In the combination in vivo experiment (Figures 1, 2 and 3), authors injected naloxone systemically (i.p.) before the oral administration of the reagents (AVFQHNCQE, Hpp11, or Captopril). Since naloxone will reach  everywhere in the body after systemic injection, it would be better to inject naloxone orally to show and expect the direct interaction between the peptides and the opioid receptors at the intestine before absorption

Response:

To ensure naloxone effect avoiding its digestion it was decided to be administered via subcutaneous to reach systemic circulation and thus, to block all opioid receptors and not only those present in the intestinal tract. This methodology has been extensively reported by other authors and that is why we considered that administration of naloxone by injection was the most accurate via of administration. To determine whether how opioid receptors blockage by naloxone could interfere in the peptide demonstrated antihypertensive effect we performed the blood pressure recordings at 0h, 2h, 4h, 6h, 8h and 24h post-peptide administration. We suggest that the peptide interact with intestinal opioid receptors because it was demonstrated that the peptide it is not absorbed and therefore, it can not be found in the systemic circulation.

Comment 4: The mechanism of the intestinal opioid-receptor mediated hypertension is not clear in this manuscript. Authors should add the description in the discussion. Will the peptides can bind to the opioid receptors?

As reported in the literature it has been demonstrated the presence of opioid receptors in the intestinal tract (Miner-Williams, Stevens, & Moughan, 2014). Our results demonstrate that the peptide was found intact and was not absorbed. In addition, in the presence of naloxone its antihypertensive effect was totally abolished. Therefore, we pointed out that the interaction with the opioid receptors present in the intestinal tract are involved in its antihypertensive effect. In addition, it has been reported that those antihypertensive peptides that produce its effect by the interaction with opioid receptors decrease blood pressure by increasing nitric oxide bioavailability. In fact, Sipola et al. demonstrated that the blood pressure-lowering effect of α-lactorphin and β-lactorphin in SHR is mediated via vasodilation in mesenteric arteries following peripheral opioid receptor stimulation and subsequent NO release (Sipola et al., 2002). Agreeing with that, the peptide AVFQHNCQE was demonstrated in a previous study to produce a nitric oxide mediated antihypertensive effect considering that in the presence of L-NAME, an inhibitor of nitric oxide (NO) synthesis, the peptide antihypertensive effect was abolished.

However, taking into consideration the complexity of heterogeneity of opioid receptor populations including different types of opioid receptors it is complex to ensure how the peptide bind the receptor. It will be highly interesting to perform in a future experiments including docking studies that will help to understand this complex interaction between our peptide and gastrointestinal opioid receptors.

Additional information has been included to clarify how peptides can bind opioid receptors (lines 62-68).

“These opioid-like peptides are atypical opioid ligands that are released from the hydrolysis of food and present N-terminal sequences which are different from those of the typical opioid peptides such as endorphins. Some common structural feature of opioid peptides is the presence of a tyrosine residue at the amino terminal end, and an aromatic residue, such as phenylalanine or tyrosine, in the third or fourth position. These structural motives seem to be necessary for binding to the opioid receptors, however there is still no consensus about critical opioid-like structures [17].”

Comment 5: If possible, authors should demonstrate the release of nitric oxide by the oral nonapeptide and the involvement of opioid receptors in the mechanism.

The nitric oxide-mediated antihypertensive effect was demonstrated in a previous study (Mas-Capdevila et al., 2019) in where in the presence of L-NAME, a nitric oxide production inhibitor, the antihypertensive effect of AVFQHNCQE was abolished.

Some authors reported that the interaction of the opioid-like peptides with the opioid receptors results in nitric oxide release and nitric oxide is one of the main vasodilators in the system (Li, Zhou, Ma, Cao, & Dong, 2013; Toda, Kishioka, Hatano, & Toda, 2009)

References:

  1. Li, M., Zhou, L., Ma, G., Cao, S., & Dong, S. (2013). The cardiovascular effects of a chimeric opioid peptide based on morphiceptin and PFRTic-NH2. Peptides, 39(1), 89–94. https://doi.org/10.1016/j.peptides.2012.10.014
  2. Mas-Capdevila, A., Iglesias-Carres, L., Arola-Arnal, A., Aragonès, G., Aleixandre, A., Bravo, F. I., & Muguerza, B. (2019). Evidence that Nitric Oxide is Involved in the Blood Pressure Lowering Effect of the Peptide AVFQHNCQE in Spontaneously Hypertensive Rats. Nutrients, 11(Cvd), 225. https://doi.org/10.3390/nu11020225
  3. Miner-Williams, W. M., Stevens, B. R., & Moughan, P. J. (2014). Are intact peptides absorbed from the healthy gut in the adult human? Nutrition Research Reviews, 27(2), 308–329. https://doi.org/10.1017/S0954422414000225
  4. Sipola, M., Finckenberg, P., Vapaatalo, H., Pihlanto-Leppälä, A., Korhonen, H., Korpela, R., & Nurminen, M.-L. (2002). Alpha-lactorphin and beta-lactorphin improve arterial function in spontaneously hypertensive rats. Life Sciences, 71(11), 1245–1253.
  5. Toda, N., Kishioka, S., Hatano, Y., & Toda, H. (2009). Modulation of Opioid Actions by Nitric Oxide Signaling. Anesthesiology, 110(1), 166–181. https://doi.org/10.1097/ALN.0b013e31819146a9

Reviewer 2 Report

Manuscript number biomolecules-839728  "Implication of opioid receptors in the 3 antihypertensive effect of a novel chicken foot-derived peptide" can be improved. In particular the results are not presented clearly, infact the rappresentation shown in the fig1 and fig 2 is not clear, it is recomended to change the display.

Indeed, even the bioavability study needs to be improved with the UHPLC-MS/MS spectra. 

Materials and methods must be integrated, it is necessary to report the  injected volume used in vivo experiments

Author Response

Manuscript ID: biomolecules-839728

Dear reviewer, Please find the point-by-point response to each of your comments with the description of the changes made in the manuscript.

Thank you very much for your attention reconsidering our manuscript for its publication in Biomolecules.

Yours sincerely,Begoña Muguerza, corresponding author.

REVIEWER 2

Comment 1: Manuscript number biomolecules-839728 "Implication of opioid receptors in the 3 antihypertensive effect of a novel chicken foot-derived peptide" can be improved. In particular, the results are not presented clearly, infact the rappresentation shown in the fig1 and fig 2 is not clear, it is recomended to change the display.

Response: As the reviewer suggested, figure 1, 2, 3 have been modified in order to improve their interpretation.

Comment 2: Indeed, even the bioavailability study needs to be improved with the UHPLC-MS/MS spectra.

Response: As explained in the manuscript, we performed a bioavailability study in plasma. However, neither the entire peptide sequence nor any of its derived fragments (predicted by the in silico and in vitro simulated gastrointestinal digestion) were found. Therefore, considering the obtained results, UHPLC-MS/MS spectra was not included. However, if the reviewer consider that it is necessary to include UHPLC-MS/MS spectra we will be pleasant to include it in the manuscript.

 Comment 3: Materials and methods must be integrated, it is necessary to report the injected volume used in vivo experiments.

Response: We appreciate the comment about the materials and methods sections but we consider that taking into account the complexity and the extensive amount of experiments the most appropriate way to explain the methodology is by following their chronologic order as explained in the results.

As the reviewer suggested we have included the injected volume used in the in vivo experiment (line 214).

“To evaluate if AVFQHNCQE presented opioid-mediated effect, SHR received via subcutaneous 100 µL of Naloxone (3 mg/kg bw), and afterwards a single oral dose (1 mL) of AVFQHNCQE (10 mg/kg bw).”

Reviewer 3 Report

The authors highlighted the antihypertensive effect of a new peptide derived from chicken feet by involving opioid receptors

The manuscript is interesting

I suggest the authors to add in the introduction  these recent manuscripts regarding the nanopeptide

Fisher AB, Dodia C, Chatterjee S, Feinstein SI. A Peptide Inhibitor of NADPH Oxidase (NOX2) Activation Markedly Decreases Mouse Lung Injury and Mortality Following Administration of Lipopolysaccharide (LPS) Int J Mol Sci. 2019;20(10):2395.

Is the peptide toxic?

The manuscript would benefit from inclusion of introducing/bridging sentences between the individual parts of the "Results" that explain the logical order and rationale for the experiments

In the conclusions , the Authors should highlight the possible clinical significance of their findings

Author Response

Manuscript ID: biomolecules-839728

Dear reviewer, Please find the point-by-point response to each of your comments with the description of the changes made in the manuscript.

Thank you very much for your attention reconsidering our manuscript for its publication in Biomolecules.

Yours sincerely,Begoña Muguerza, corresponding author.

REVIEWER 3

Comment 1: The authors highlighted the antihypertensive effect of a new peptide derived from chicken feet by involving opioid receptors. The manuscript is interesting. I suggest the authors to add in the introduction these recent manuscripts regarding the nanopeptide.

Fisher AB, Dodia C, Chatterjee S, Feinstein SI. A Peptide Inhibitor of NADPH Oxidase (NOX2) Activation Markedly Decreases Mouse Lung Injury and Mortality Following Administration of Lipopolysaccharide (LPS) Int J Mol Sci. 2019;20(10):2395.

Response:

We appreciate the reviewer suggestion about including recent manuscripts and we have included some new references about food derived peptides and their relation with opioid receptors. Unfortunately, the manuscript “A Peptide Inhibitor of NADPH Oxidase (NOX2) Activation Markedly Decreases Mouse Lung Injury and Mortality Following Administration of Lipopolysaccharide (LPS)” was not included considering that it does not fit into the manuscript topic.  

Comment 2: Is the peptide toxic?

Response:

The peptide AVFQHNCQE was identified from Hpp11 a chicken foot protein hydrolysate. Both Hpp11 and AVFQHNCQE were extensively employed in in vitro and in vivo experiments and in any case was observed any toxic effects. In addition, considering the potential antihypertensive effects of Hpp11 and AVFQHNCQE, clinic studies are being raised and these will include toxicological study. However, both Hpp11 and the peptide are obtained from chicken foot protein which is an edible food matrix. This fact and the results we have so far suggests that it will present any toxic effect.

Comment 3: The manuscript would benefit from inclusion of introducing/bridging sentences between the individual parts of the "Results" that explain the logical order and rationale for the experiments.

Response:

Considering the reviewer comments, we have performed some changes to improve result and discussion section which will help to explain the experiment steps followed in the present manuscript.

Comment 4: In the conclusions , the authors should highlight the possible clinical significance of their findings

Response:

Considering the reviewer comments, we included the following sentence about the possible clinical significance of the present findings. 

“The results presented in this work that includes the mechanistic study of the way of action of AVFQHNCQE are relevant because open doors to use of this chicken-foot derived peptide to formulate antihypertensive functional foods or nutraceuticals that could allow to treat prehypertension or to prevent the development of hypertension.”

Round 2

Reviewer 1 Report

Authors appropriately responded to all my comments and the revised version of the manuscript is fine for the publication.